# Simultaneous Analysis of 53 Pesticides in Safflower (*Carthamus tinctorius* L.) by Using LC–MS/MS Coupled with a Modified QuEChERS Technique

**DOI:** 10.3390/toxics11060537

**Published:** 2023-06-16

**Authors:** Wei Song, Chuanyi Peng, Yuxin Liu, Fang Han, Haitao Zhu, Dianbing Zhou, Yu Wang, Lijun Chen, Xiaodi Meng, Ruyan Hou

**Affiliations:** 1Technical Center for Hefei Customs, Hefei 230022, China; 2Anhui Province Key Laboratory of Analysis and Detection for Food Safety, Technical Center for Hefei Customs, Hefei 230022, China; pcy0917@ahau.edu.cn (C.P.);; 3Key Laboratory of Food Nutrition and Safety, School of Tea and Food Science & Technology, Anhui Agricultural University, Hefei 230036, China

**Keywords:** g-C_3_N_4_, modified QuEChERS, liquid chromatography–tandem mass spectrometry, pesticide residues, safflower

## Abstract

Objective: An optimized quick, easy, cheap, effective, rugged, and safe (QuEChERS) technique was investigated and compared with the conventional QuEChERS technique for the simultaneous analysis of fifty-three pesticide residues in safflower using ultra-high performance liquid chromatography–tandem mass spectrometry (UHPLC–MS/MS). Method: Graphitic carbon nitride (g-C_3_N_4_) consisting of a major amount of carbon and nitrogen with a large surface area was used as a QuEChERS adsorbent instead of graphitized carbon black (GCB) for safflower extraction purification. Validation experiments were performed using spiked pesticide samples, and real samples were analyzed. Results: The linearity of the modified QuEChERS technique was evaluated with high coefficients of determination (R-2) being higher than 0.99. The limits of detection were <10 μg/kg. The spiked recoveries ranged from 70.4% to 97.6% with a relative standard deviation of less than 10.0%. The fifty-three pesticides exhibited negligible matrix effects (<20%). Thiamethoxam, acetamiprid, metolachlor, and difenoconazole were detected in real samples using an established method. Conclusion: This work provides a new g-C_3_N_4_-based modified QuEChERS technique for multi-pesticide residue analysis in complex food matrices.

## 1. Introduction

Plants are often subjected to biotic stress, especially insect and disease attacks [1] Pesticides are necessary and have been routinely used in agriculture production for decades to protect crops against pests and diseases, but their widespread use also brings environmental pollution and potential risks to human health via the food chain. In order to reduce pesticide damage, more and more pesticides are becoming restricted or forbidden during the planting process through legal directives [1,2], and new environmentally friendly bio-pesticides with low toxicity and high efficiency have been developed. At present, pesticides in plant-derived foods should be monitored regularly to ensure the residues do not present a danger to the public. Developing a convenient, sensitive, and highly repeatable method for detecting multi-pesticide residues is vital and welcomed.

Safflower (*Carthamus tinctorius* L.) is a traditional Chinese medicine with a homology of medicine and food. It is widely planted in many regions in the world and naturally contains a very large amount of flavonoids, fatty acids, pigments, volatile oils, and other active ingredients, which exhibit various physiological activities, including antibacterial [3], anti-inflammatory [3], analgesic [4], antioxidant [5,6,7], and antitumor activities [8]. In view of these active ingredients, safflower has the potential to be used as a raw material to prepare bioactive foods [9], but it also results in a very complex food matrix for the determination of pesticide residues [10,11]. It is well known that a complex matrix is not conducive to the analysis of target analytes. Furthermore, a complex matrix, if there is no effective pretreatment and purification, can also cause damage to the chromatographic separation system and even diminish the mass spectrometer sensitivity for target analytes. Consequently, the minimization of matrix effects and developing a simultaneous qualitative and quantitative method for multi-pesticide residues in safflowers is critical in reducing food safety risks. The rapid, easy, cheap, effective, rugged, and safe (QuEChERS) methods are modern sample preparation techniques with good reproducibility, satisfactory relative recoveries, and very low relative standard deviation (RSD values) in the pesticide multiresidue analysis of various matrices [12,13,14]. The conventional QuEChERS technique generally uses octadecylsilane (C_18_), primary secondary amine (PSA), and graphitized carbon black (GCB) as clean-up materials [15] and has been employed. However, the application of the conventional QuEChERS technique on all plant-derived food samples with different complex matrix interferences is limited and unavailable [16]. Therefore, a modified QuEChERS towards target analytes in specific food matrices is vital and necessary.

Graphitic carbon nitride (g-C_3_N_4_), a two-dimensional non-metallic layered material, consists of a major amount of carbon and nitrogen with a large surface area, which is similar to graphene [17]. g-C_3_N_4_ possesses an attractive electronic structure, and this structure has advantages in the field of chemical and thermal stability and is nontoxic, has low cost and density, and synthesizes easily [18,19,20]. The chemical stability, optical properties, and biocompatibility of g-C_3_N_4_ enable it to be combined with a wide range of materials and to have many functions [20,21,22]. In particular, the sp2-hybridized nitrogen atoms in the tri-s-triazine structure of g-C_3_N_4_, featuring a lone pair of electrons, improve the overall adsorption of organic molecules [23,24]. Liu et al. successfully prepared Fe_3_O_4_-g-C_3_N_4_-PDA@MIL-101 and exhibited the great adsorption efficiency of three kinds of organophosphorus pesticides in green onions and cabbage [17]. ZnO/g-C_3_N_4_ nanoflowers were synthesized and exhibited promising performance in the simultaneous analysis of pesticide residues in different sample matrices (tea, water, pear, and cucumber) [25]. To date, the application of g-C_3_N_4_ employed as a QuEChERS adsorbent to minimize matrix effects in multi-pesticide residue analysis is limited.

The pesticides selected in this study are commonly used against insect and disease attacks in the process of agricultural production, and the key monitoring targets of market supervision and pesticides are easily exceeded. The matrix of safflower is complicated, which is not conducive to pesticide residue analysis. In this study, g-C_3_N_4_ was used as a QuEChERS adsorbent for minimizing the interfering substances effect in the matrix of safflower. Thus, this work aimed at developing an effective modified QuEChERS technique coupled with ultra-high-performance liquid chromatography–tandem mass spectrometry (UHPLC–MS/MS, American TEDIA Co., Fairfield, OH 45014, USA) for the simultaneous analysis of 53 pesticide residues in the complex food matrix of safflower.

## 2. Materials and Methods

### 2.1. Reagents and Materials

The fifty-three pesticide standards (purity 98.0–99.9%) selected for this study were obtained from Dr. Ehrenstorfer Company GmbH (Augsburg, Germany). The acetonitrile and methanol (mass spectrometry grades) used as the mobile phase in the UHPLC–MS/MS analysis were obtained from American TEDIA Co., Ltd. All of the tested aqueous solutions were prepared with deionized water with 18.2 MΩ cm resistivity (Millipore company, Boston, MA, USA). The formic acid (purity ≥ 99.0%) and ammonium acetate (purity 99.0%) used as the aqueous phase were obtained from Sigma-Aldrich (St. Louis, MO, USA). Urea (99%), ethanol, anhydrous magnesium sulfate, sodium chloride, and anhydrous sodium acetate were purchased from Shanghai Sinopharm Group (China). PSA, GCB, and C_18_ sorbent were obtained from Anpel Laboratory Technologies Inc. (Shanghai, China). The safflower derived from Xinjiang, China, were randomly obtained from local supermarkets and tested as real samples in the present study.

### 2.2. Preparation and Characterization of g-C_3_N_4_

In the current study, g-C_3_N_4_ was prepared with some modifications [17,25]. Briefly, urea was gradually heated to 80 °C at a rate of 5 °C min^−1^; this temperature was maintained for 2 h. The urea was finally heated to 580 °C at the same rate, and maintained for another 2 h. After the calcination process, the powder was ultrasonically dispersed in ethanol for 4 h, and afterwards dried to obtain g-C_3_N_4_. An X-ray diffraction (XRD) study was performed at room temperature to determine the crystallinity using a SmartLab (9) diffractometer (Rigaku Corporation, Tokyo, Japan). To obtain the Fourier-transform infrared (FTIR) spectra (500–4000 cm^−1^), infrared absorption was characterized using a Nicolet8700 spectrometer (Nicolet, Madison, WI, USA). The surface morphology was analyzed using a Talos F200X transmission electron microscope (FEI, Hillsboro, OR, USA) coupled with an EDXRF-8100 energy-dispersive X-ray spectrometer (Shimadzu, Kyoto, Japan).

### 2.3. Safflower Sample Pretreatment

The safflowers were ground for pretreatment. First, 1 g safflower sample was weighed and subjected to static extraction for 15 min with pure water (2 mL) in a 50 mL centrifuge tube, and then acetonitrile (10 mL) was added. The sample was allowed to mix thoroughly by vortexing for 2 min; 2 g NaCl was added subsequently, and the sample was vortexed for another 2 min and then subjected to high-speed centrifugation at 8000 rpm for 5 min. The supernatant (1 mL) sample was transferred into a polypropylene centrifuge tube and further cleaned up using a modified QuEChERS technique. After strong shaking manually and mixing thoroughly by gently vortexing for 2 min, the solution was centrifuged at 8000 rpm for 5 min. The supernatant was collected and diluted twice with deionized water and then filtered through a 0.22-μm filter membrane. The filtrate was collected and stored at −20 °C until ready for UHPLC–MS/MS analysis.

### 2.4. UHPLC–MS/MS Analysis

The samples were analyzed using a UHPLC–MS/MS system (1290–6460, Agilent, USA). Sample separation was carried out on a ZORBAX SB-C_18_ chromatographic column (2.1 × 100 mm, 3.5 μm). During UHPLC, the mobile phase consisted of 5 mM ammonium acetate/0.1% formic acid (*w*/*v*, A) and acetonitrile (B), and the detailed gradient elution was 1–30% B for 0–3 min; 30–40% B for 3–6 min; 40% B for 6–9 min; 40–60% B for 9–15 min; 60–90% B for 15–19 min; and 90% B for 19–23 min. The parameters were set as follows: flow rate, 0.4 mL/min; column temperature, 40 °C; volume of injection, 10 μL.

Mass spectrometry was conducted as follows: atomizer pressure of 280 kpa, drying gas temperature of 300 °C, drying gas speed of 10 L/min, capillary voltage of 4000 V, and dynamic multiple reaction monitoring scanning mode. In the positive ion mode, the qualitative ion, quantitative ion, fragmentation voltage (FP), and collision energies (CE) are listed in Appendix A.

### 2.5. Statistical Analysis

Each sample was determined in triplicate, and values were expressed as the mean ± standard deviation (SD). The matrix effects (MEs) of the fifty-three targeted pesticides in this study were calculated according to Meng et al., Rutkowska, Lozowicka, and Kaczynski, and Lee et al. [10,26,27].

## 3. Results 

### 3.1. Characterization of g-C_3_N_4_

The crystal structure of g-C_3_N_4_ synthesized material was investigated using X-ray diffraction, as shown in Figure 1a. The X-ray diffraction pattern exhibits two distinct peaks at 13.01° and 27.85° that are assigned to the planes of (100) and (002), respectively accroding to Wang et al. [28]. The characteristic one at 27.85° demonstrates the stacking of aromatic, which is indexed for graphitic material [28]. The Fourier-transform infrared spectra of g-C_3_N_4_ is shown in Figure 1b. The FTIR peak at 814 cm^−1^ provides evidence of the existence of g-C_3_N_4_ with tri-s-triazine rings. The absorption peaks between 1230 and 1650 cm^−1^ can be attributed to C–N and C=N stretching vibrations [29]. 

The morphological structure of g-C_3_N_4_ was analyzed using high-resolution transmission electron microscopy. Irregular agglomerated particles and thicker areas are shown in Figure 2a,b, which indicate the presence of stacking layers. The results of elemental mapping in Figure 2c further indicate the uniform distribution of C and N. The EDS spectrum confirms the presence of abundant C and N in the g-C_3_N_4_ (Figure 2d).

### 3.2. UHPLC–MS/MS Method Development

#### 3.2.1. Optimization of UHPLC–MS/MS Parameters

In total, 53 pesticides of different categories were selected as target analytes for UHPLC-MS/MS, including benzimidazole fungicides, organophosphorus insecticides, carbamates, triazoles, herbicides, and neonicotinoid insecticides. All 53 target analytes formed high-abundance [M + H]^+^ parent ions in the positive electrospray ionization mode. Appendix A presents the qualitative and quantitative ions and the optimized collision energy values for each pesticide. The extracted ion chromatography of 53 pesticides is shown in Figure 3, and the retention time is also summarized in Appendix A.

#### 3.2.2. Optimization of the QuEChERS Methodology

The extract solvent affects extraction efficiency. Acetonitrile, the most commonly used organic reagent in the QuEChERS technique, is used to extract various pesticides, covering a wide range of polarities of pesticide residues in several plant foodstuffs, unlike extract solvents such as acetone and ethyl acetate. Acetone is highly water soluble; it cannot be easily separated from water. Ethyl acetate is ineffective as an extract solvent for polar pesticides. In this study, acetonitrile effectively and consistently extracted 53 pesticides from safflower; therefore, acetonitrile was selected as the extract for the subsequent experiments.

In theory, NaCl affects the extract solvent polarity, but improves the selectivity, and MgSO_4_ shows favorable water absorption properties and greatly facilitates the release of acetonitrile [30]. In the experiment, NaCl yielded higher detection rates and observed signal-to-noise ratios for 53 pesticides than did MgSO_4_.

Further purification of the extracted solution is required to minimize the drawbacks arising from the complex matrices. In particular, the matrix components, some of which are colored, are present in large amounts and interfere with the pesticide residue analysis. In the present study, g-C_3_N_4_ was used to remove interfering substances from the matrix safflower extract instead of GCB. g-C_3_N_4_ had obvious adsorption effects on the matrix of safflower extract (Figure 4) with a great adsorption efficiency. The applicability of the modified QuEChERS technique (150 mg MgSO_4_, 50 mg PSA, 50 mg C_18_, and 100 mg g-C_3_N_4_) was initially compared with the performance from the conventional QuEChERS technique (150 mg MgSO_4_, 50 mg PSA, 50 mg C_18_, and 25 mg GCB). The recoveries of 53 pesticides in safflower with two QuEChERS techniques are shown in Figure 5.

Obviously, the recovery target compounds of the conventional QuEChERS technique were less effective (2.0–98.3%) than that of the modified QuEChERS technique: the recoveries of some pesticides, i.e., carbendazim, thiabendazole, pyrimethanil, chlorantraniliprole, cyprodinil, aldicarb, and coumaphos, were even less than 5.0%. Herein, the conventional QuEChERS technique could not meet the minimum requirements of multi-pesticide residue analysis. The modified QuEChERS technique removed pigments effectively and recovered all target compounds satisfactorily (61.7–100.0%), and the recoveries for the major pesticides ranged from 80.0% to 100.0%. These results demonstrated that g-C_3_N_4_, which removes interfering substances without the adsorption of target compounds, is a suitable pretreatment material during the analysis of pesticide residues in safflower. Therefore, the modified QuEChERS technique provided clean-up performances and was selected for further experiments.

Sample dilution during the extraction and purification steps has been shown to reduce matrix component interference in the analysis of broccoli, tea, and apple [12,31], but the dilution can also reduce the overall sensitivity of the instrument. In the present study, the purified solution was diluted twice with water before injection. The degree of dilution maintained a sufficient compromise between the removal of matrix component interference and analysis sensitivity.

### 3.3. Analytical Performance and Verification

The method established in this study was validated by evaluating the linearity, recovery, limit of detection (LOD), limit of quantification (LOQ), and precision under optimum conditions for fifty-three pesticides (Table 1). The correlation coefficients of the fifty-three target pesticides were 0.9976–1.0000 (Appendix A). The LOD (signal-to-noise ratio = 3) was 0.3–3.0 μg/kg, and the LOQ (signal-to-noise ratio = 10) was 1–10 μg/kg. To evaluate the accuracy (expressed as recovery %) of the developed method, the safflower samples were spiked with fifty-three pesticides at 10 μg/kg, 20 μg/kg, and 100 μg/kg, respectively. As shown in Table 1, the mean recoveries ranged from 70.4% to 97.6%. The precision values described as relative standard deviation (RSD) were lower than 10%.

### 3.4. Matrix Effect Analysis

The presence of coextracted compounds affects the ionization of target compounds; thus, the matrix effect influences the quality of quantitative data obtained using UPLC–MS/MS by calculating a matrix effect coefficient [12,31,32]. A value between −20% and 20% suggests a weak matrix effect; a value between −50% and 20% suggests a medium matrix effect; and a value that is greater than −50% or 50% suggests a strong matrix effect. On the basis of the coefficients obtained from this study, the matrix effects ranged from −2.9% to −10.1% (Table 1), indicating that the fifty-three pesticide residues had weak matrix effects.

### 3.5. Monitoring 53 Pesticides in Real Samples

The analysis of 53 pesticide residues in 10 safflower samples was achieved using the modified QuEChERS technique. Thiamethoxam was detected in five safflowers ranging from 12.1 to 46.7 μg/kg. Acetamiprid was detected in three safflowers ranging from 6.7 to 28.3 μg/kg. Both metolachlor and difenoconazole were detected in only one safflower at a concentration of 5.7 μg/kg and 50.7 μg/kg, respectively.

## 4. Discussion

The crystal structure of g-C_3_N_4_ synthesized in this study is consistent with previous studies [19,28]; however, it is completely different from the XRD pattern of commercial GCB [28,33] as well as the morphological structure. As we know, commercial GCB is always opaque and steel-gray to black with a metallic luster and a greasy feel, presented as hexagonal crystals or thin leaf-like layers. Unlike GCB, g-C_3_N_4_ has a six-membered carboatomic ring structure, similar to N-doped carbon materials, and has a tri-s-triazine structure composed of carbon and nitrogen elements. Importantly, g-C_3_N_4_ has no obvious adsorption effect on nonpolar pesticides and planar structure target compounds. Due to the presence of –NH– and –NH_2_ groups, g-C_3_N_4_ has more electrons and surface groups than carbon materials [28]. Alkaline sites, electron-rich properties, and hydrogen bonds of g-C_3_N_4_ facilitate the adsorption of interfering substances in saffron and other pigments in the safflower extract.

As for the liquid–liquid phase separation, the desolvation efficiency is closely related to the mobile phase [34]. The composition and separation behavior of the mobile phase influence the MS ionization performance [35]. In this study, acetonitrile was investigated and used as the organic phase, which can reduce baseline interference and increase the responses for most pesticides. The use of an acidic solution in the aqueous phase improved the ionization efficiency and the intensity of the target compounds. The application of ammonium acetate was beneficial to the symmetry of the target compounds. These aforementioned assumptions have been confirmed by the findings of previous studies [10,36].

Complex matrices usually produce serious interference that affects the accuracy of analysis. Therefore, effective pretreatment is required. The QuEChERS technique is a common pretreatment method in pesticide residue analysis. Key parameters in the QuEChERS methodology include extract solvent, partition salt, and clean-up adsorbent. Acetonitrile is not affected by interference from lipophilic compounds, lipids, and pigments [37]. NaCl and MgSO_4_ are two commonly used salts for liquid–liquid phase separation [37]. In accordance with previous research findings, NaCl generally yielded a higher detection rate than did MgSO_4_, possibly because NaCl removes highly water-soluble polar matrix interference.

For the conventional QuEChERS technique, C_18_, PSA, and GCB are the three adsorbents commonly used to remove interfering substances from food. C_18_ has been shown to retain minerals, fats, and vitamins, whereas PSA removes many polar organic acids, sugars, fatty acids, and polar pigments from the extract [30]. This is because the amine group of PSA and the polar matrix can form hydrogen bonds through weak ion exchange. GCB is often used to eliminate sterols and pigments from plant-derived samples; however, GCB adsorbs nonpolar pesticides and substances with planar functionality [38,39], which has a significant influence on the recovery of compounds with hexahedral structures through π–π interactions [12,38,39].

A comparative evaluation was conducted to assess the proposed QuEChERS technique with recently reported methods for multi-pesticide residue analysis in different food matrices. As presented in Table 2, the modified QuEChERS technique in this study was effective and provided a highly beneficial effect on multi-pesticide residue analysis in safflower.

## 5. Conclusions

In this study, a modified QuEChERS technique coupled with UHPLC–MS/MS for fifty-three pesticide residues in safflower was proposed. g-C_3_N_4_ was synthesized, characterized, and was considered an effective adsorbent to minimize the matrix effect in safflower. In comparison with the conventional GCB-based QuEChERS technique, the g-C_3_N_4_-based modified QuEChERS technique presented satisfactory recovery performance, as well as low LODs and LOQs, indicating that the proposed method has high accuracy, precision, and reproducibility. In view of the presented results, the developed QuEChERS technique is very suitable for the accurate qualitative and quantitative target multi-pesticide residue analysis in complex food matrices. Herein, further study will focus on expanding the scope of complex food matrices for multi-residue pesticide analysis, such as tea.

## Figures and Tables

**Figure 1 toxics-11-00537-f001:**
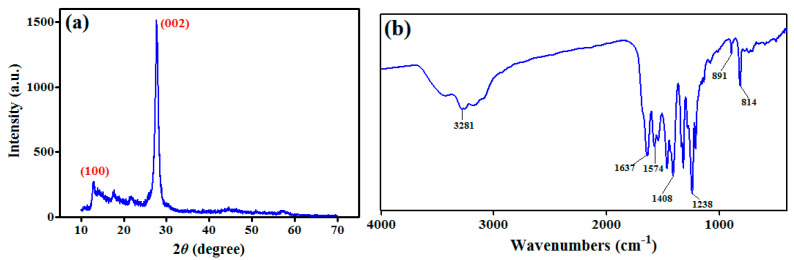
X-ray diffraction pattern (**a**) and Fourier-transform infrared spectra (**b**) of g-C_3_N_4_.

**Figure 2 toxics-11-00537-f002:**
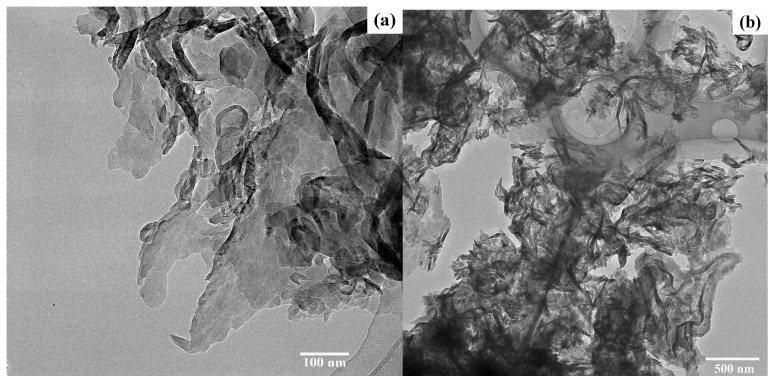
High-resolution transmission electron microscope images of (**a**) g-C_3_N_4_ (100 nm), (**b**) g-C_3_N_4_ (500 nm), (**c**) mapping of g-C_3_N_4_, and (**d**) EDS of g-C_3_N_4_.

**Figure 3 toxics-11-00537-f003:**
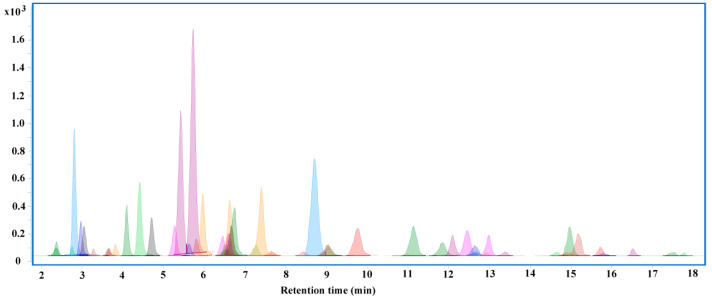
Characteristic ion extraction chromatogram of 53 pesticides.

**Figure 4 toxics-11-00537-f004:**
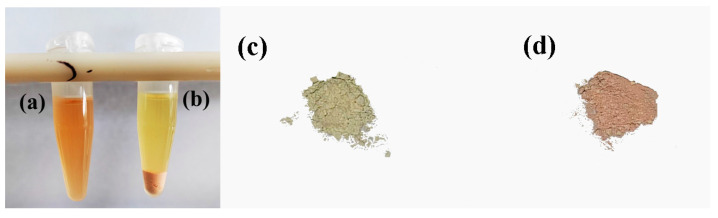
Purification effect of g-C_3_N_4_. (**a**) Original safflower extraction, (**b**) safflower extraction after adsorption, (**c**) original g-C_3_N_4_ powder, and (**d**) g-C_3_N_4_ powder after adsorption.

**Figure 5 toxics-11-00537-f005:**
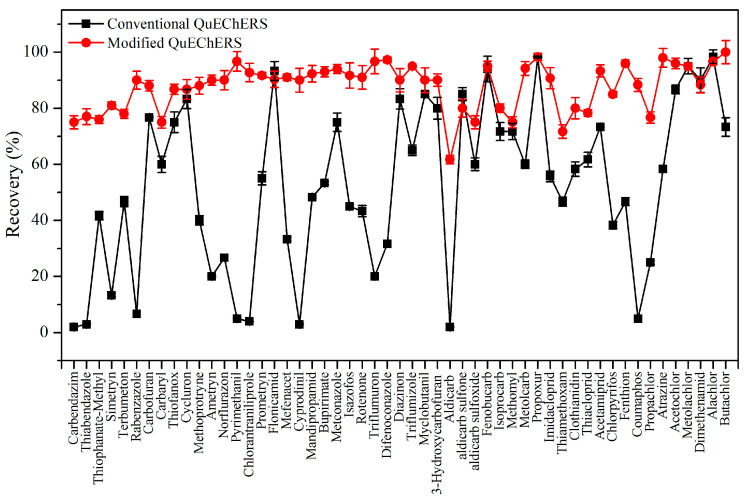
Influence of different QuEChERS techniques on the recoveries of 53 pesticides in safflower.

**Table 1 toxics-11-00537-t001:** Mean recovery for 53 pesticide residues in safflower.

Pesticides	Spiked 10 μg/kg		Spiked 20 μg/kg		Spiked 100 μg/kg		LOD	LOQ	ME
	Mean Recovery (%)	RSD (*n* = 3) (%)	Mean Recovery (%)	RSD (*n* = 3) (%)	Mean Recovery (%)	RSD (*n* = 3) (%)	(μg/kg)	(μg/kg)	(%)
Carbendazim	75.4 ± 5.4	7.1	78.5 ± 5.1	6.5	92.8 ± 3.9	4.2	0.3	1	−9.5
Thiabendazole	77.2 ± 5.7	7.4	82.0 ± 5.5	6.7	91.4 ± 4.6	5.0	0.3	1	−8.1
Thiophanate-Methyl	77.4 ± 6.7	8.6	78.3 ± 5.8	7.4	87.4 ± 3.8	4.3	1.5	5	−9.7
Simetryn	81.8 ± 6.7	8.2	87.9 ± 8.3	9.4	93.6 ± 4.8	5.1	0.6	2	−7.8
Terbumeton	76.9 ± 4.2	5.4	79.4 ± 4.1	5.2	91.5 ± 3.8	4.1	0.6	2	−9.3
Rabenzazole	86.0 ± 7.1	8.2	88.1 ± 5.9	6.7	94.4 ± 5.0	5.3	3	10	−5.2
Carbofuran	87.0 ± 7.5	8.6	86.8 ± 7.1	8.2	97.1 ± 6.3	6.5	3	10	−6.1
Carbaryl	79.9 ± 6.6	8.2	81.4 ± 6.4	7.9	93.2 ± 4.8	5.2	3	10	−9.5
Thiofanox	84.8 ± 7.6	9	87.7 ± 5.5	6.3	95.3 ± 4.2	4.4	3	10	−6.3
Cycluron	83.2 ± 5.2	6.2	87.0 ± 5.1	5.9	97.2 ± 3.7	3.8	0.6	2	−6.7
Methoprotryne	75.7 ± 6.3	8.3	79.1 ± 6.5	8.2	90.2 ± 5.7	6.3	1.5	5	−6.2
Ametryn	83.2 ± 7.2	8.7	87.2 ± 7.1	8.1	95.7 ± 5.4	5.6	0.6	2	−5.3
Norflurazon	79.4 ± 7.7	9.7	81.3 ± 7.0	8.6	90.2 ± 6.0	6.7	0.6	2	−5.6
Pyrimethanil	88.7 ± 7.4	8.3	89.7 ± 6.4	7.1	93.9 ± 5.2	5.5	3	10	−3.7
Chlorantraniliprole	86.1 ± 5.7	6.6	88.2 ± 5.4	6.1	93.9 ± 4.2	4.5	3	10	−4.6
Prometryn	83.5 ± 7.6	9.1	88.8 ± 7.4	8.3	94.1 ± 6.1	6.5	0.3	1	−4.2
Flonicamid	87.5 ± 6.6	7.5	89.4 ± 6.3	7.0	95.7 ± 5.6	5.9	3	10	−4.9
Mefenacet	86.9 ± 7.0	8.1	89.8 ± 6.3	7.0	96.7 ± 5.4	5.6	1.5	5	−4.5
Cyprodinil	80.3 ± 6.9	8.6	81.1 ± 6.6	8.1	94.9 ± 5.9	6.2	0.6	2	−5.7
Mandipropamid	81.9 ± 5.3	6.5	84.4 ± 5.2	6.2	94.2 ± 4.1	4.3	0.6	2	−3.8
Bupirimate	88.4 ± 6.3	7.1	85.5 ± 5.6	6.5	96.7 ± 5.4	3.9	1.5	5	−3.5
Metconazole	85.9 ± 7.3	8.5	88.6 ± 7.3	8.2	96.8 ± 5.7	5.9	3	10	−3.1
Isazofos	81.1 ± 6.3	7.8	84.1 ± 5.8	6.9	93.9 ± 3.8	4.0	0.6	2	−4.2
Rotenone	81.0 ± 5.9	7.3	84.8 ± 5.9	6.9	95.9 ± 3.9	4.1	3	10	−4.5
Triflumuron	82.0 ± 7.2	8.8	82.2 ± 7.2	8.8	94.3 ± 4.0	4.2	3	10	−3.7
Difenoconazole	87.9 ± 7.8	8.9	88.2 ± 7.2	8.2	96.1 ± 5.0	5.2	3	10	−3.3
Diazinon	79.8 ± 6.1	7.6	81.8 ± 5.6	6.8	95.6 ± 4.2	4.4	1.5	5	−5.1
Triflumizole	86.6 ± 7.2	8.3	87.0 ± 6.5	7.5	97.1 ± 5.0	5.1	0.3	1	−3.5
Myclobutanil	82.6 ± 6.0	7.3	85.5 ± 5.6	6.6	96.0 ± 4.9	5.1	0.6	2	−5.2
3-Hydroxycarbofuran	80.3 ± 7.1	8.9	81.2 ± 6.3	7.7	95.1 ± 5.9	6.2	3	10	−5.5
Aldicarb	78.3 ± 7.2	9.2	77.7 ± 6.6	8.5	89.0 ± 6.0	6.7	3	10	−9.2
Aldicarb sulfone	76.3 ± 6.8	8.9	78.8 ± 6.5	8.3	87.2 ± 5.3	6.1	3	10	−8.0
Aldicarb sulfoxide	77.1 ± 7.5	9.7	84.1 ± 7.1	8.5	90.4 ± 5.7	6.3	3	10	−9.5
Fenobucarb	80.6 ± 5.2	6.4	84.4 ± 4.8	5.7	93.6 ± 4.0	4.3	1.5	5	−4.5
Isoprocarb	81.3 ± 6.9	8.5	80.9 ± 6.8	8.4	93.3 ± 6.2	6.6	3	10	−8.3
Methomyl	75.8 ± 6.7	8.9	78.6 ± 5.6	7.1	92.5 ± 3.3	3.6	3	10	−9.8
Metolcarb	87.1 ± 7.1	8.2	88.9 ± 7.4	8.3	96.5 ± 5.3	5.5	3	10	−2.9
Propoxur	86.6 ± 7.4	8.6	87.8 ± 6.8	7.8	97.4 ± 5.9	6.1	3	10	−3.8
Imidacloprid	86.8 ± 7.3	8.4	89.2 ± 6.6	7.4	97.6 ± 4.5	4.6	3	10	−4.7
Thiamethoxam	70.4 ± 6.1	8.7	76.9 ± 6.1	7.9	92.7 ± 5.7	6.1	3	10	−10.1
Clothianidin	78.8 ± 6.9	8.8	80.5 ± 7.0	8.7	94.3 ± 5.8	6.2	3	10	−9.7
Thiacloprid	76.8 ± 6.3	8.2	80.6 ± 6.1	7.6	94.9 ± 5.4	5.7	0.6	2	−9.8
Acetamiprid	86.9 ± 6.7	7.7	87.6 ± 6.8	7.8	95.3 ± 5.2	5.5	1.5	5	−3.3
Chlorpyrifos	78.3 ± 6.4	8.2	81.8 ± 6.9	8.4	91.3 ± 5.4	5.9	3	10	−7.5
Fenthion	88.5 ± 6.5	7.4	88.9 ± 6.4	7.2	94.1 ± 4.3	4.6	3	10	−3.8
Coumaphos	78.2 ± 6.3	8.1	81.8 ± 5.9	7.2	90.6 ± 4.8	5.3	3	10	−5.8
Propachlor	73.9 ± 5.0	6.8	76.3 ± 4.7	6.1	91.7 ± 4.6	5.0	0.6	2	−9.7
Atrazine	85.8 ± 6.8	7.9	88.3 ± 6.1	6.9	92.9 ± 4.2	4.5	1.5	5	−3.0
Acetochlor	83.3 ± 6.6	7.9	85.9 ± 6.3	7.3	93.6 ± 4.8	5.1	3	10	−4.3
Metolachlor	87.8 ± 6.6	7.5	87.5 ± 4.9	5.6	96.9 ± 4.4	4.5	0.6	2	−3.9
Dimethenamid	75.0 ± 5.7	7.6	79.6 ± 5.3	6.7	90.7 ± 4.2	4.6	0.6	2	−5.8
Alachlor	85.9 ± 7.5	8.7	88.9 ± 6.8	7.6	93.4 ± 4.8	5.1	3	10	−4.7
Butachlor	82.9 ± 5.1	6.2	85.2 ± 5.5	6.4	92.3 ± 3.6	3.9	3	10	−4.6

**Table 2 toxics-11-00537-t002:** Comparison of the proposed QuEChERS technique with recently reported methods for multi-pesticide residue analysis.

Clean-Up Method	Sample Matrix	Quantity of Pesticides	LOD (μg/kg)	LOQ (μg/kg)	Time (min)	Recovery (%)	Reference
Without Clean-Up	Wheat Flour	37	/	10	18	70–120	[40]
MgSO_4_ + PSA + C_18_	Rice and Wheat	100	/	3.6	27.4	70–116	[41]
MgSO_4_ + Multi-Walled Carbon Nanotubes	Cowpea	65	5–10	/	9	70–120	[42]
MgSO_4_ + PSA + C_18_ + GCB	Golden Berry	42	1.5	5	8	70–114	[43]
MgSO_4_ + PSA + GCB	Lettuce	16	/	5	/	70–120	[44]
MgSO_4_ + PSA + C_18_ + Carb	Fruits and Vegetables	54	0.003–2	0.01–6.67	16	73.2–134.3	[45]
MgSO_4_ + PSA	Mango	68	0.5–7	2–25	5	70–122	[46]
MgSO_4_ + PSA + C_18_ + Florisil	Pear	170	/	2.5–10	/	70–120	[47]
MgSO_4_ + PSA + C_18_ + GCB	Tobacco	55	8–23	25–75	10 + 35	63–161	[48]
MgSO_4_ + C_18_ + ChloroFiltr	Green Vegetables	164	/	5	20	70–120	[49]
MgSO_4_ + PSA + C_18_ + GCB	Beef	129	/	0.003–11.37	/	70.5–128.1	[35]
MgSO_4_ + PVPP + PSA + GCB	Tea	134	/	<10	23	66.8–118.3	[10]
MgSO_4_ + PVPP + PSA + GCB	Polyphenol-Rich Foods	20	/	10–20	/	73–106	[12]
Sodium Acetate+Ammonium Acetate/Na_2_-EDTA + PSA + C_18_	Fish and Shrimp	66	<5	<10	18	70–125	[16]
MgSO_4_ + PSA + C_18_ + GCB	Safflower	53	/	/	18	2.0–98.3	This work
MgSO_4_ + PSA + C_18_ + g-C_3_N_4_	Safflower	53	0.3–3	1–10	18	70.4–97.6	This work

## Data Availability

Not applicable.

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
