# Peer review of "Simultaneous Analysis of 53 Pesticides in Safflower (Carthamus tinctorius L.) by Using LC–MS/MS Coupled with a Modified QuEChERS Technique"

_toxics, 2023, doi:10.3390/toxics11060537_

Round 1

Reviewer 1 Report

This paper describes an adsorbent for eliminating the interfering substances effect in the matrix of safflower and a evaluation between the QuEChERS and modified QuEChERS extraction.

The title should include the scientific name of safflower.

The abstract should present the context of the study. What is g-C3N4?

This article was submitted to the special issue Quality Control and Safety Management of Tea. From my point of view this article is out of the scope, please describe the relationship between the special issue and the content of this article.

To prove the novelty and importance of the study it is essential to contextualize it with more recent literature. I recommend that the majority of all cited works should be less than five years. The introduction present 12 references before 2018. Please update all the sections.

The introduction section should include some statics of pesticide residues in Safflower at least in China.

Why is important the Safflower? worldwide or only in China? the relevance of this study and the justification is scare in the introduction section.

There is no justification about to choose 53 pesticides. Why are relevant? are relevant as a pesticide residue? in food? environment?

All the calculations, analytical performance and verification should be described in details in the methodology.

The methodology should describe the origin of real samples, how were sampled, why only 10 samples? are representative? etc

After to read this article, the section Results and Discussion should be separated.

Authors can build a discussion to help readers to follow the logical development of the manuscript. I invite authors to build the arguments in favor of how this work contributes to theory and provides directions to experts.

Conclusion should not be a summary of your study or an extension of the discussion of results. The presented work has some merits, but at the same time, the provided future research directions appear to be trivial.

Author Response

Dear reviewer,

We are very grateful to you and your positive comments and insightful critiques. We have revised the manuscript according to your great  comments, please find the detailed response in the attachment.

All the other modifications in revised manuscript were marked with red changes.

Reviewer 2 Report

The manuscript by Wei et al. entitled “Elimination of Matrix Effect and Simultaneous Detection of 53 Pesticides in Safflower Through LC–MS/MS Coupled with a Modified QuEChERS Technique” is attempting to establish a simultaneous multi-residue analysis of fifty-three pesticides in Safflower. The topic is interesting, but the arrangement of results is not satisfactory! Thus, the following specific comments and suggestions should be addressed to modify the manuscript's quality:

Title: Revise the title. In particular, replace “Elimination” with any suitable word. The matrix effect can be minimized or reduced by using modified QuEChERS, but complete elimination is not possible for analytical chemistry. Revise the word “Detection” with “Analysis”.

Abstract: The abstract should contain objectives, methods, results, and conclusions. Please revise and rearrange the abstract to reflect the sequence. Add novelty of this study.

Introduction: Revise the last paragraph of the Introduction, and mention the novelty of this study. Replace “detection” with “analysis”. Explain the specific objective in detail.

“However, the application of the conventional QuEChERS technique on all plant-derived food samples with different complex matrix interference is limited (Shin, Kim, & Kang, 2021).”  Replace published year from 2021 to 2015.

Materials and method: There is a lacking of “Statistical analysis”. Please add this section in the method and material section. In addition, a detailed illustration of sample pretreatment and analytical instrument should be provided either in the main text or as a supplementary file.

Figures:

For Figures 1 and 2, it should be better to add a comparative characterization of GCB and gC3N4. A comparison in illustration will be more vivid and logical to establish your study design.

Figures 3 and 4 literally have no significant meaning without a comparison of GCB and gC3N4 effect should be presented. The chromatographic peak differences between original QuEChERS and modified QuEChERS should be a wondrous option. Only the gC3N4 effect without comparison in these figures is highly discouraged. Please revise those figures.

The author should follow the following papers add cite those for matrix effect description:

1.      https://doi.org/10.1016/j.foodchem.2018.11.130

2.      https://doi.org/10.3390/foods11192930

3.      https://doi.org/10.3390/foods12061226

Figure 5 is also difficult to follow, authors are advised to follow the above-mentioned papers for a clear presentation of matrix effect results during multi-residue pesticide analysis.

Table 1: Please present the recovery mean (mean value ± Standard error) in the table.

Reference: Follow the MDPI journal (toxics) style.

Conclusion: It is too short. Please elaborate on the conclusion by adding key findings and recommendations for further studies.

I think that English correction by a native speaker will be better.

Author Response

(The authors gave the same response as above.)

Round 2

Reviewer 1 Report

Dear authors

The paper is suitable for publication,

Best Regards

Reviewer 2 Report

I think that the authors have corrected the contents I pointed out well.